# Machine Learning on Visibility Graph Features Discriminates the Cognitive Event-Related Potentials of Patients with Early Alzheimer’s Disease from Healthy Aging

**DOI:** 10.3390/brainsci13050770

**Published:** 2023-05-07

**Authors:** Jesse Zhang, Jiangyi Xia, Xin Liu, John Olichney

**Affiliations:** 1Computer Science Department, University of Southern California, Los Angeles, CA 90089, USA; jessez@usc.edu; 2UC Davis Center for Mind and Brain, Davis, CA 95618, USA; jixia@ucdavis.edu; 3UC Davis Computer Science Department, Davis, CA 95616, USA; xinliu@ucdavis.edu

**Keywords:** machine learning, Alzheimer’s Disease, EEG, visibility graph, event-related potential, mild cognitive impairment, prodromal, electroencephalography

## Abstract

We present a framework for electroencephalography (EEG)-based classification between patients with Alzheimer’s Disease (AD) and robust normal elderly (RNE) via a graph theory approach using visibility graphs (VGs). This EEG VG approach is motivated by research that has demonstrated differences between patients with early stage AD and RNE using various features of EEG oscillations or cognitive event-related potentials (ERPs). In the present study, EEG signals recorded during a word repetition experiment were wavelet decomposed into 5 sub-bands (δ,θ,α,β,γ). The raw and band-specific signals were then converted to VGs for analysis. Twelve graph features were tested for differences between the AD and RNE groups, and *t*-tests employed for feature selection. The selected features were then tested for classification using traditional machine learning and deep learning algorithms, achieving a classification accuracy of 100% with linear and non-linear classifiers. We further demonstrated that the same features can be generalized to the classification of mild cognitive impairment (MCI) converters, i.e., prodromal AD, against RNE with a maximum accuracy of 92.5%. Code is released online to allow others to test and reuse this framework.

## 1. Introduction

There is mounting evidence suggesting that AD may be primarily a synaptic disorder [1] and synaptic abnormalities occur before any clinical symptoms. EEG measures instantaneous excitatory and inhibitory postsynaptic potentials [2], and thus provides a powerful non-invasive tool to capture synaptic dysfunction underlying very early cognitive changes in AD. The superior temporal resolution of EEG makes it especially advantageous in detecting changes in complex multi-stage cognitive processes such as memory, a key indicator of early AD [3]. A large number of studies have demonstrated that EEG measures, including event-related potentials (ERPs) and oscillations, are sensitive to subtle brain changes in early AD [4,5,6]. Applying a word repetition paradigm, designed to elicit brain activity related to language and memory processing, our laboratory has identified several ERP/oscillatory measures that reliably distinguish mild cognitive impairment (MCI) and early-stage AD patients from healthy elderly controls [7,8,9,10,11,12]. For example, our ERP studies revealed that the N400 component, sensitive to semantic processing and integration, and the P600 (or ‘Late Positive Component’, LPC), sensitive to explicit verbal memory, can be reliably elicited in healthy elderly but not in MCI or AD patients [7,8,9,13]. In mild AD, both the N400 and the P600 word repetition effects are diminished [13], whereas MCI and preclinical AD patients show compromised P600 but relatively preserved N400 effects [7,8,9]. Similarly, our EEG oscillatory analyses revealed a power suppression in the alpha range (9–11 Hz) that is attenuated for repeated relative to new words in healthy elderly [10]. This alpha word repetition effect is also compromised in amnestic MCI and correlated with verbal memory measures [10].

A limitation of traditional ERP/oscillatory analyses is that they usually focus on the timing and the magnitude of pre-defined components at the expense of the overall pattern and complexity of EEG data. Some prior works convert EEG signals to visibility graphs (VGs) [14], which preserve many features of the original EEG signal. Converting resting state EEG signals to VGs allows for discriminative graph features to be discovered [15] and utilized in high-accuracy neural network based classification (98%) between AD patients and normal elderly [16].

Other studies which have applied neural networks or other machine learning algorithms to resting state EEG in AD include Morabito et al. [17], who used convolutional neural networks on 19 channel EEG and achieved a three-class AD/MCI/cognitively normal (CN) classification accuracy of  82% [17,18]. Zhao and He [19] combined deep belief networks with support vector machines on 16 channel EEG signals and achieved  92% accuracy classifying AD vs. CN [18,19]. Duan et al. [20] quantified between-channel connectivity of resting-state EEG signals in MCI and mild AD patients using coherence measures; they used the Resnet-18 model [21] to classify between MCI and controls, and AD and controls with an average 93% and 98.5% accuracy, respectively.

Despite the promise of the above studies and other machine learning algorithms which have used biomarkers of AD to improve diagnostic accuracy [22], there are still to date no widely used machine learning algorithms for the clinical diagnosis of AD. Historically, clinical diagnosis of possible and probable AD (generally found to be between 80 and 90% accurate in clinicopathological studies) was based on recognizing the typical cognitive and behavioral symptoms of this dementia and the exclusion of other possible causes of dementia, whereas a “definite AD” diagnosis was only possible via invasive brain measures from a biopsy or autopsy providing histopathological evidence of AD [23]. Currently, the International Working Group (IWG) recommends that the clinical diagnosis of AD be restricted to those with positive biomarkers together with specific AD phenotypes [24]. While purely biological definitions of AD (e.g., [25]) have become more widely used for research purposes in recent years, the IWG considers the present limitations of biomarkers sufficient that they should not be used for the diagnosis of disease in the cognitively unimpaired [24]. Thus, the “gold standard” for the clinical diagnosis of AD is criteria (e.g., [23,26]) which incorporate multiple biomarkers (including markers of amyloid-β (Aβ) and tau pathology, neuronal injury and neurodegeneration) along with the clinical phenotype. With the rapid emergence of machine learning algorithms into medical research, this could, however, change rapidly in upcoming years [27].

Our hypothesis is that word repetition tasks, which have been shown sensitive to detect MCI-to-AD conversion and even preclinical AD using ERPs [7,8,11], can also be used to discriminate AD from normal elderly with high accuracy using a VG-based machine learning approach. Compared to resting state EEG, word repetition task signals are expected to yield better discriminative features given that verbal memory impairments are the best predictors of MCI to AD conversion [28]. Combining these two lines of past work, we converted EEG signals recorded during word repetition experiments to visibility graphs. We operated under the assumption that the ERP components of interest will be preserved after conversion to graphs and features extracted from these graphs will encode the ERP components while reducing variance across subject data for better downstream machine learning classification performance.

Therefore, this work focuses on the analysis of EEG signals and extracting features from them that are useful in discriminating between AD and RNE in a variety of machine learning algorithms. To demonstrate the generalizability of those features, we tested whether they can also effectively discriminate between prodromal Alzheimer’s (pAD, MCI patients who converted to Alzheimer’s Dementia within 3 years) and robust normal elderly (RNE, normal elderly persons who have remained cognitively normal for the duration of follow-up). We apply a similar approach to that of Ahmadlou et al. [16], although extracting many more features (including many novel ones in this context) from word repetition task EEG signals (instead of resting state as in Ahmadlou et al. [16]).

In our framework, pictured in Figure 1, we first collect EEG data from word repetition tasks. We then perform pre-processing of this data and then convert the EEG signals to visibility graphs. From these VGs we extract 12 features and perform statistical tests for feature selection, keeping the discovered statistically significant predictors as inputs for machine learning algorithms. Finally, the dimensionality of this feature space is reduced with principal component analysis and we use the resulting reduced feature space as inputs to machine learning algorithms.

In summary, the intended contributions of this work are threefold:We demonstrate the effectiveness of EEG analysis on *word repetition tasks* for dementia classification (AD vs. RNE) across various machine learning algorithms (support vector machines, logistic regression, linear discriminant analysis, neural networks);We select a new set of high performing features under a framework for EEG visibility graph analysis that, when combined with existing features from the literature, detect even earlier stage AD (i.e., discriminates pAD vs. RNE);We open source our code so that it can be adapted for other datasets and tasks (e.g., resting state EEG data or discriminating other types of dementia).

## 2. Methods

Figure 1 details the framework. Open-sourced code for our method is available online at https://github.com/jesbu1/ML-Visibility-Graphs-for-Alzheimers (accessed on 19 March 2023).

### 2.1. Participants

EEG and behavioral data were taken from 15 patients with probable AD (mean age 78.5 years, range 67–91) [23] recruited primarily through the Alzheimer’s Disease Research Centers at the University of California, San Diego and the University of California, Davis. Additional data were taken from 15 patients with amnestic MCI (mean age 74.6 years, range 60–84) [29] who later converted to dementia and 11 healthy elderly controls (mean age 74.1 years, range 57–79) who were recruited in a previous published longitudinal study [8]. See Table 1 for participant details. All participants were screened for treatable causes of cognitive impairments such as vitamin B12 deficiency and thyroid dysfunction, and underwent a brain scan (generally MRI) prior to enrollment. The exclusion criteria included stroke, epilepsy and psychiatric conditions, as well as several classes of central nervous system (CNS) active medications.

The patients were tested with an EEG word repetition paradigm and clinical assessments. At the initial baseline recording session, the 15 MCI patients all met Petersen Criteria for amnestic MCI [31] but not for dementia [32]. Probable AD was diagnosed according to criteria set out by the National Institute of Neurological and Communicative Disorders and Stroke–Alzheimer’s Disease and Related Disorders Association [23]. The 15 MCI patients subsequently converted to AD within 3 years of their initial baseline session (mean number of years 1.62 ± 0.7). In the present study we focus on the initial baseline ERP data in order to investigate neural activity associated with AD and prodromal AD (pAD, MCI to AD conversion within 3 years). For more information about participant demographics and their neurocognitive test results please refer to [7,10].

### 2.2. Word Repetition Paradigm

For each trial, participants were presented with an auditory phrase describing a category (e.g., “a type of wood”, “a breakfast food”) followed by a visually presented target word ∼1 s later (presentation duration = 0.3 s, visual angle ∼= 0.4${}^{\circ }$). The target words were nouns, which were either congruous (e.g., ‘cedar’) or incongruous with the preceding category phrase with a probability of 0.5. Congruous and incongruous words were matched on the frequency of usage (mean = 32, SD = 48; [33]) and word length (mean of 5.8 characters, SD = 1.6) [12]. Participants were instructed to wait for 3s after the onset of each target word, read the word aloud and then give a yes/no decision indicating whether the word fit the preceding category. No time limit was imposed on making responses. Of all the category–word pairs, 1/3 only appeared once, 1/3 appeared twice and the other 1/3 appeared 3 times (congruous and incongruous pairs were counterbalanced). For those items that appeared twice, the lag between the first and the second presentation was short (0–3 intervening trials, spanning ∼10–40 s). For those items that appeared 3 times, the lags between presentations were longer (10–13 intervening trials, spanning ∼100–140 s). A total of 432 trials were performed in 3 blocks. The six word conditions tested include All New (AN), New Congruous (NC), New Incongruous (NI), All Old (AO), Old Congruous (OC) and Old Incongruous (OI). Further details of the experimental paradigm have been published previously [8,12].

### 2.3. EEG Signal Preparation

Across participants, EEG was recorded from 19 to 32 channels including midline (Fz, Cz, Pz) and lateral (F7/F8, T5/T6, O1/O2) sites in the International 10–20 System and additional sites approximate Broca area (Bl/Br), Wernicke area (Wl/Wr) and Brodmann area 41 (L41/R41). EEG signals were recorded with a 250 Hz sampling rate, bandpassed between 0.016 and 100 Hz, and re-referenced offline to averaged mastoids. Data preprocessing and artifact rejection were performed using MATLAB [34] with EEGLAB [35] and Fieldtrip [36] toolboxes. EEG epochs were extracted and time-locked to the onset of target words, 2 s before and 2 s after the word onset, and visually inspected for non-physiological artifacts. Independent component analysis [37] was then applied to isolate and remove eye movement artifacts. Artifact-removed EEG epochs were then mirror-padded to 8 s (2 s to the beginning and 2 s to the end) and bandpass filtered into five frequency bands (δ 1–4 Hz, θ 4–8 Hz, α 8–13 Hz, β 13–30 Hz, γ 30–45 Hz), using zero-phase Hamming-windowed sync finite impulse response filters as implemented in the EEGLAB (pop_eegfiltnew). This function automatically selects the optimal filter order and transition bandwidth to minimize filter distortions and maximize time precision. For each of the five frequency bands of interest, a high-pass filter was first applied and then a low-pass filter. Transition bandwidths were set to be 25% of the passband edge for passband edges >4 Hz, with −6 dB cutoff frequency at the center of the transition band. For the 4 Hz passband we used a transition bandwidth of 2 Hz and for the 1 Hz passband (δ band) we used a transition bandwidth of 1 Hz. Finally, raw and bandpass filtered EEG segments were extracted 1 s before and 2 s after the word onset for further analyses.

### 2.4. Time Series to Visibility Graph Conversion

For every patient, we obtained multiple word repetition trials for each experimental condition. To reduce the noise in the EEG signal and extract event-related information, we averaged across trials in each condition so that there was one averaged EEG time series per condition, frequency band and channel combination for each patient. Each time series was then averaged into 80 ms non-overlapping epochs (the values of every 20 timesteps were averaged together). This was done for three reasons:Reduce the amount of computing time required for data analysis.Reduce the amount of variance within the individual EEG signals to prevent the machine learning models overfitting to the data. We can think of this process acting like a low-pass filter, helping reduce signal noise from muscle artifacts commonly present in frequencies above 12 Hz.Reduce the variance across participants’ data when performing hypothesis testing as a result of reducing the variance within their signals.

All time series were finally shortened to 1 s pre-stimulus to 2 s post-stimulus.

### 2.5. Visibility Graphs (VG)

Visibility graphs (VG), first proposed by Lacasa et al. [14], inherit many properties of the time series they represent. For example, a VG corresponding to a periodic series will be regular and one corresponding to a random series will be random. VGs were first utilized in EEG analysis in a paper by Ahmadlou et al. [16] in order to classify Alzheimer’s patients against RNE with a classification accuracy of 97.8%.

Intuitively, a visibility graph of a time series **x** is created by considering each *i*’th point of the time series and determining which other time points are visible from it. The *i*’th node of the VG is connected with an undirected edge to any nodes visible from it. Formally, two nodes of the VG, $a_{m}$ and $a_{n}$, are connected with an undirected, weight 1 edge if and only if:$$x_{m+j}<x_{n}+\left(\frac{n-(m+j)}{n-m}\right)\left(x_{m}-x_{n}\right)\;\;\;\;\forall j\in Z_{+}\;:j<n-m$$

Figure 2 demonstrates the creation of a VG. The top graph represents the original time series, while the graph underneath represents the corresponding nodes and edges of the visibility graph. There is a line connecting points in the time series (and an undirected edge in the corresponding VG) if and only if those two points are visible from each other. Visibility graphs allow for features to be extracted which can encode temporal locality (as a node is always connected to its direct neighbors in the original EEG signal) but also features which capture information from nodes that are farther away, as nodes that are visible from each other will be connected, even if they are far away in time in the original signal. In general, VGs are biased towards creating local edges that capture information about the signal over short periods of time, with the exception of peaks in the signal. VGs can also only be extracted per electrode; however, we compensate for this by also extracting a cross-channel feature, as detailed below.

### 2.6. Feature Extraction

In total, we used 12 features to classify the Alzheimer’s (AD) and the robust normal elderly (RNE) groups. Six of these features have been tested in previous EEG graph theory studies of AD, namely clustering coefficient sequence similarity [15], average clustering coefficient [38,39], global efficiency [15,38,40], local efficiency [15,41], small-worldness [15] and graph index complexity [15,16]. The other six are graph features heavily studied in the field of computer science that, to the best of our knowledge, have not yet been considered in EEG graph theory studies. In general, the new features we introduce come from classic, well-studied problems in graph theory and are targeted towards extracting information specifically about VG structure (e.g., visibility of vertices induced by the time series structure). Every feature is extracted from each condition, band and channel combination, and then compared across groups with a two-tailed *t*-test. The entire feature extraction process was performed in Python 3 using Numpy [42], Scipy [43] and NetworkX [44], three open source packages that were essential for data formatting, *t*-testing and graph analysis, respectively. In all definitions below, |V| denotes the number of vertices in the graph and |E| denotes the number of edges.

#### 2.6.1. Clustering Coefficient Sequence Similarity (CCSS)

All visibility graphs are constructed from a single time series (a single-channel EEG signal), making it easy to compare individual time series across groups. Visibility graph similarity, proposed by Ahmadlou et al. [45] and modified for EEG VG analysis by Wang et al. [15], is a method of comparing the similarity of multiple time series across groups by measuring the similarity of the nodes’ degree sequences in the VGs. As suggested by Wang et al. [15], the similarity of clustering coefficients is utilized instead of degree sequences to generate connections between the visibility graphs of different channels under a single band-condition combination [15]. Networks are generated by making each channel a node and connecting an edge between two nodes if the CCSS between their VGs is above a certain threshold, θ, which was chosen to be 0.25 based on Wang’s results and our own empirical data. We note that this is the only all-channel comparison measure and employs the functional network concept, while all subsequent features are based on single-channel visibility graph features. CCSS between two VGs *X* and *Y* is calculated as follows:(1)$$CCSS=|\frac{cov(CCS(X),CCS(Y))}{\sigma_{CCS(X)}\sigma_{CCS(Y)}}|$$
where CCS is the “clustering coefficient sequence,” or sequence of clustering coefficients where each clustering coefficient measures how close the vertex’s neighbors are to becoming a complete graph (clique) [46]. The clustering coefficient *C* of a node *i* is defined as:(2)$$C_{i}=\frac{2|E_{i}|}{|K_{i}\left|(\right|K_{i}\left|-1\right)}$$
where $|E_{i}|$ denotes the number of edges of the neighbors of a node *i*, $|K_{i}|$ indicates the number of neighbors of node *i* and $\frac{|K_{i}\left|(\right|K_{i}\left|-1\right)}{2}$ represents the number of possible connections in a complete graph consisting of node *i*’s neighbors. During *t*-testing, we compared the average number of edges per person between the two groups.

#### 2.6.2. Average Clustering Coefficient

The average clustering coefficient is defined as simply the average of clustering coefficients defined in Equation (Equation 2).
(3)$$C=\frac{1}{\left|V\right|}\sum_{i=1}^{\left|V\right|}C_{i}$$

The average clustering coefficient measures the average tendency of neighbors of nodes to become complete graphs. In context, it denotes the likelihood of our EEG signals to be shaped in a way that allows for close interconnectedness in the VG.

#### 2.6.3. Global Efficiency

Global efficiency is defined as the average of the inverse shortest path lengths between all nodes. The shortest path length $d_{ij}$ between two nodes in our VG construction, *i* and *j*, is defined to be the minimum number of edges needed to traverse from *i* to *j* or *j* to *i*. Thus, global efficiency, $E_{global}$, is defined as
(4)$$E_{global}=\frac{1}{\left|V|(|V|-1\right)}\sum_{i,j,i\neq j}\frac{1}{d_{ij}}$$

It is interpreted as sum of all inverse shortest path distances divided by the number of shortest path distances counted. A higher global efficiency corresponds to a network that is more efficient at transmitting/combining information and relates to the small-worldness of the network [15,38,47,48,49,50]. In context, a higher global efficiency in a VG means that there are likely more EEG time points that are visible from other points which are relatively farther away in time.

#### 2.6.4. Local Efficiency

The local efficiency of a graph is the average of the global efficiencies of each subgraph composed of every vertex’s direct neighbors. It is similar to the average clustering coefficient; however, during its calculation vertices outside of each subgraph can be taken into account in the shortest path between two nodes [15]. Local efficiency, $E_{local}$, is defined as
(5)$$E_{local}=\frac{1}{\left|V\right|}\sum_{i}\frac{1}{|V_{g_{i}}\left|(\right|V_{g_{i}}\left|-1\right)}\sum_{j,k,j\neq k}\frac{1}{d_{jk}}$$
where $|V_{g_{i}}|$ represents the number of vertices in the subgraph of vertex *i* (composed only of its direct neighbors) and |V| represents the number of vertices of the entire graph [47]. As each edge in our VG is of weight one, a higher local efficiency corresponds to more direct edges on average in each subgraph, indicating EEG signals with variations in voltage that allow for a greater number of direct connections between points close in time.

#### 2.6.5. Small-Worldness

Small-worldness is a measure of how much a graph acts like a small-world network. Small-world networks have the property that the typical distance between any two randomly chosen vertices grows logarithmically in terms of total number of vertices of a graph [47]. As logarithmic functions grow very slowly, this correlates with low average shortest path lengths and high global efficiencies and clustering coefficients. A measure of small-worldness, *S* was defined by Humphries and Gurney [51] as
$$S=\frac{C/C_{r}}{L/L_{r}}$$
where $C,C_{r}$ are the average clustering coefficients of the graph in question and a random graph, respectively, and $L,L_{r}$ are the average shortest paths lengths between all pairs of vertices in the graph in question and the random graph, respectively. Our random graphs were generated with the Erdös–Rényi method [52], and the same random graph was used to compare all VGs.

#### 2.6.6. Graph Index Complexity (GIC)

GIC, proposed by Kim and Wilhelm [53], is a measure of graph complexity. It is defined as
(6)GIC=4c(1−c)
where
(7)$$c=\frac{\lambda_{max}-2\cos(\pi /(|V|+1))}{\left|V|-1-2\cos(\pi /(|V|+1)\right)}$$
$\lambda_{max}$ represents the largest eigenvalue of the adjacency matrix of the graph. This eigenvalue lies somewhere between the average and max node degree. Therefore, a larger GIC may correspond to a more complex signal structure resulting from, for example, more frequent signal voltage fluctuations.

#### 2.6.7. Size of Max Clique

A clique is a subset of vertices of a graph such that they form a complete subgraph—all vertices have direct edges to each other [54]. Therefore a maximum clique is just the clique with the largest number of vertices in the graph. As clique-finding in graph theory is known to be in a class of problems that may always take exponential time to solve, a fast deterministic approximation algorithm that, in the worst case, overestimates by a factor proportional to $\left|V\right|/{\left(\log|V|\right)}^{2}$ was used [55]. Max clique was selected by the authors as a feature because it can account for a specific cluster of time points in the EEG signal that are shaped differently across groups, leading to a complete subgraph in the VG of differing numbers of vertices.

#### 2.6.8. Cost of Traveling Salesman Problem (TSP)

The traveling salesman problem asks the question: what is the shortest cost tour in a graph that starts from a vertex, visits all other vertices in the graph and then returns to the starting vertex [54]? This problem is also difficult for computers to solve efficiently [56]; therefore, a deterministic approximation algorithm that overestimates by at most a factor of 2 was implemented [54]. As all edge weights in our VGs are 1, this essentially amounts to the shortest length tour that visits all vertices, starting and ending at the vertex that corresponds to the first time point of the EEG. The TSP path cost provides another measure of graph complexity that can signify significant differences in EEG wave structure across groups.

#### 2.6.9. Density

Graph density is a measure of how close a graph is to having the maximum number of edges. It is simply the actual number of edges divided by the maximum possible number of edges [57]. Density, *D*, for an undirected graph is defined as
(8)$$D=\frac{2|E|}{\left|V|(|V|-1\right)}$$
as it can have at most $\frac{\left|V|(|V|-1\right)}{2}$ edges. Density can highlight differences in the number of edges of VGs across groups.

#### 2.6.10. Independence Number

The independence number is the size of the largest independent set of a graph, which is the largest set of vertices such that no two vertices share an edge [58]. This can be reduced to the max clique problem [55]; therefore, a similar approximation algorithm was used to determine the independence number. A higher independence number could indicate an EEG signal shape that allows for more, or different, timepoints to be invisible from each other.

#### 2.6.11. Size of Minimum Cut

In this context, the minimum cut is defined to be a partition of the vertices into two disjoint sets such that the number of edges across the cut is minimized. This feature was analyzed because a difference in minimum cut size across the two groups could indicate timepoints in the EEG signal that are on average more or less visible (therefore having differing numbers of edges) from other vertices.

#### 2.6.12. Vertex Coloring Number

Vertex coloring describes the problem of finding the minimum number of colors required to color a graph such that no two vertices that share an edge have the same color. We used a deterministic approximation algorithm that colors vertices in order from largest to smallest degree as the problem is extremely difficult to exactly solve computationally [59]. The number of colors required to color a graph is likely to be different between two graphs if there is a significant difference in EEG signal structure.

## 3. Statistical Analysis and Feature Selection

As stated in *Feature Extraction*, we utilize a two-tailed *t*-test, as implemented in the open-source library Scipy [43], to determine statistical significance in graph features between groups. A *p*-value of 0.01 was determined as the threshold for significance.

### PCA

We select all features with a *p*-value of less than 0.01. A high number of feature combinations combined with a false positive rate of 1% lead us to use principal component analysis (PCA), a method of linearly mapping features from a higher dimensional space onto a lower dimensional subspace spanned by the eigenvectors that account for the directions of highest variance. As suggested by Ahmadlou et al. [16], we apply PCA to reduce the dimensionality of the feature space to about 10% of its original dimensionality. At a high level, PCA can be interpreted as a way to linearly project the vector onto a lower-dimensional latent space such that the distance between the original and projected latent datapoints is minimized.

Specifically, the “principal components” of PCA are calculated by performing an eigendecomposition of the covariance matrix of the data. Consider an n×d matrix X where *n* is the number of datapoints (number of patients in our study) and *d* is the dimensionality of the feature space (the statistically significant features discovered by *t*-testing). The input matrix X is first normalized, and then the principal components and their weights can be discovered by the following eigendecomposition:(9)$$\frac{1}{n-1}XX^{T}=\underset{\mathrm{components}}{\underset{︸}{P}}\;\underset{\mathrm{component}\;\mathrm{magnitudes}}{\underset{︸}{D}}\;\underset{\mathrm{components}}{\underset{︸}{P^{T}}}.$$
Assuming the prinicpal component vectors and their magnitudes are sorted in descending order of magnitude, then feature reduction to *k* features is performed by taking the first *k* components. These *k* components thereby intuitively correspond to the *k* directions of highest variance in the data. In our study we use PCA to reduce the data dimensionality to 11, selected from cross-validation of values around 10% of the number of original features.

## 4. Machine Learning Classifiers

We test a variety of machine learning algorithms for classification: linear logistic regression, linear soft-margin support vector machines (SVM), linear discriminant analysis (LDA) and a fully connected artificial neural network (ANN). The first three algorithms are chosen to test linear separability; logistic regression and support vector machines are widely used in the literature, and LDA operates under intuitive statistical assumptions about the class distributions being Gaussian and having the same covariance matrices. The neural network is chosen to approximate more complicated decision boundaries. We utilize a simple, two-layer, fully connected neural network with ReLU activations because, after feature extraction, the input is no longer temporally or spatially dependent (this excludes the use of recurrent networks or convolutional networks).

We detail each class of machine learning algorithms below:Logistic regression: logistic regression is a commonly used, simple classifier that learns a linear decision boundary by learning a single feature vector through gradient descent.Support vector machines: SVMs learn a decision boundary with a “margin” away from datapoints from either class that is maximized. This can result in better testing error as the decision boundary should not lie too close to points of either class.Linear discriminant analysis: LDA models the data distributions of both classes as Gaussians with equal covariances and draws a linear decision boundary between the means of the two Gaussians. LDA can perform very well if the input data follows these assumptions.Artificial neural network: the ANN can linearize non-linear decision boundaries in the feature space of the input data. It has the potential to overfit more easily to the data but also to learn better-fit decision boundaries if the true decision boundary must be non-linear.

For all algorithms and all comparisons (AD vs. RNE, pAD vs. RNE), the features extracted come from the AD vs. RNE comparison.

## 5. Results

We found 72 statistically significant (p<0.01) features. The total number of features tested was 5976 (resulting in 60 features that are expected to be false positives). The total 5976 is derived from 15 channels × ((5 bands + 1 raw) × 11 single-channel features × 6 conditions) + ((5 bands + 1 raw) × 1 all-channel feature (CCSS) × 6 conditions)). We utilize PCA to reduce the number of features down to 11, close to the number of features expected to be true positives, in order to combat the high number of expected false positives.

### 5.1. Statistical Analysis

All features except for CCSS showed up at least once as a significant discriminator between our subject groups. Furthermore, every band (δ,θ,α,β,γ) and the raw signal produced discriminating features. Table 2 compares the number of features produced by each condition (the number of expected false positives for each condition is 10) and band combination. The word conditions are defined in the *Word Repetition Paradigm* subsection. Additionally, Table 3 displays the number of features produced by each channel for all conditions.

While all sub-bands and the raw signal produced at least one significant result, the δ and α sub-bands and raw signal seems to be the most effective in discriminating across groups. As an example, Figure 3 demonstrates the mapping from the average EEG time series for each group for the raw signal in electrode R41 under the condition Old Congruous to averaged node degree time series (i.e., the number of other timepoints visible from each timepoint in the voltage graph).

Finally, we visualize the separation of patient classes by projecting the 72 features down to 2 dimensions in Figure 4. In the comparison of all patient classes against each other (top), we see three clear clusters of points for each class. Notably, the pAD group lies in between the AD and RNE groups in the top subfigure. This may be because the features extracted from the AD vs. RNE comparison are likely less significant for the pAD patients, although they are still general enough to create clear separation between the three groups. In the AD vs. RNE projection plot (bottom right), we see that all classifiers are able to perfectly separate the two groups, even in two dimensions. The pAD vs. RNE plot (bottom left) also demonstrates very good separability between the two groups in two dimensions, although the two sets of points are closer together than in the AD vs. RNE comparison.

The 10 most important features for each two-dimensional PCA projection comparison from Figure 4 are listed in Table 4. The feature–band–electrode combinations that were shared across at least two PCA comparisons are bolded and numbered in the tables. The α and δ bands produced the largest number of these shared combinations, and the most common features in these were global efficiency, density, TSP and GIC in electrodes F8, Fz, O1 and R41. In every single shared combination, the value of the feature increased in the groups with dementia. The increase in these feature values generally indicates an increase in the number of edges between nodes, indicating significantly different ERP structure that results in the change in their VGs.

### 5.2. Classification

All models are trained and evaluated 100 times, each time randomly splitting the dataset into a training set of 85% of the patients and a testing set composed of the remaining 15%. We report classification metrics on the 15% testing set, where the metrics are averaged across all 100 trials for each model. The PCA feature reduction step is performed each time only on the features for the patients in the training set. The best results across all classifiers are obtained by reducing the dimension of the feature vector to 11 via PCA. On the AD vs. RNE comparison, we utilize the features extracted on the full dataset for classification to test the ability of the features discovered and analyzed in Section 5.1. To test the generalization of these features, we use *only the 72 discovered AD vs. RNE features* for classification of the pAD patient group.

In summary, the accuracy for AD vs. RNE was 100% across all classifiers and the best discrimination of pAD vs. RNE—using the 72 discovered AD vs. RNE features to measure generalization—was 92.5% with the ANN. Notably, AD vs. RNE was perfectly solved with linear classifiers and pAD vs. RNE classification performance with the same classifiers was also excellent. Logistic regression achieved perfect precision and a very high AUC score (0.99); however, the ANN provided even higher accuracy (92.5 %) and very similar AUC and precision scores. Table 5 presents the classification metrics (accuracy, precision, recall, AUC).

We perform an additional comparison with K-fold cross-validation in Appendix A, with similar classification results across the board. We also examine the effect of the six novel features we introduce by reporting classification results using only the six VG features from prior work as input to the machine learning models; accuracy on the pAD vs. RNE comparison drops for every model, demonstrating the importance of the new VG features we introduce. See Appendix A for more details.

## 6. Discussion and Conclusions

This EEG/ERP word repetition paradigm has been shown to be sensitive to MCI and the conversion from MCI to AD [7,8,10]. The recent development of VGs for EEG allows for a more holistic measure of EEG time series using graph features. By combining VGs with the EEG word repetition paradigm, we are able to discriminate AD from RNE with a perfect accuracy of 100% using linear classifiers and generalize these same features for pAD vs. RNE classification with an accuracy of 92.5%—on par with previous work directly comparing pAD and RNE [60,61,62,63,64,65,66]. Our analysis demonstrates the effectiveness of looking at word repetition EEG tasks for the features we selected for this visibility graph approach.

A number of graph features including GIC, global efficiency, clustering coefficient, small-worldness and local efficiency were already confirmed to be significant in some band–electrode combinations in resting state EEG VG studies comparing Alzheimer’s to RNE [15,16]. Our results extend these findings by showing that these features also discriminate AD from RNE using a word repetition task EEG paradigm. Novel features introduced in this paper have been shown to encode more differences between AD and RNE in word repetition trials. To minimize type I error, we utilized PCA to reduce the number of input metrics used for classification. Both novel and previously studied features appeared in the top two components of our PCA loading table (Table 4). The most common features were global efficiency, density, TSP cost and GIC. Two of these features, namely TSP and density, are from the six novel ones we introduced. The presence of these features generally points to a difference in EEG time series structure between groups, especially with regards to voltage differences and overall structure differences in the waveforms. We note that min cut size, max clique size and independence number also appear in Table 4, indicating that five out of six of the novel features we introduced are important for prediction.

Learned graph features, representing group differences in the morphology of EEG time series, may reflect AD pathological changes in the neural generators of ERPs, including N400 and P600. Putative N400 generators have been found in the anterior fusiform gyri and other temporal cortical regions [67,68]. The primary neural generators of the P600 word repetition effect were localized by functional MRI to the hippocampus, parahippocampal gyrus, cingulate, left inferior parietal cortex and inferior frontal gyrus [69,70]. Extended synaptic failure in these regions due to AD pathology may account for the N400 and P600 abnormalities in AD and prodromal AD patients. For example, abnormal memory-related P600 may be associated with tau load in the medial temporal lobe (MTL), including the hippocampus, entorhinal and perirhinal cortices, based on the evidence that early tau accumulation in these regions correlates with lower memory performance and reductions in functional connectivity between the MTL and cortical memory systems [71].

Using raw and bandpass filtered EEG data, we find that the δ band produced the largest number of features, closely followed by the α band. Neural oscillations in different frequency bands are thought to carry different spatial and temporal dimensions of brain integration. Spatially, slow oscillations integrate large neural networks whereas fast oscillations synchronize local networks [72]. Temporally, slow neural fluctuations are related to the accumulation of information over long timescales across higher order cortical regions [73]. In line with these hypotheses, empirical evidence has indicated that slow oscillations in the delta range are important for higher cognitive functions that require large-scale information integration (see Güntekin and Başar [74] for a review). Delta activity has been shown to play important roles in language comprehension such as chunking words into meaningful syntactic phrases [75]. Slow wave activity (SWA) also facilitates memory consolidation during sleep by orchestrating fast oscillations across multiple brain regions [76]. It may therefore be hypothesized that cognitive impairments in AD are related to alterations in slow oscillatory activity. Accumulating evidence has supported this hypothesis, showing that decreased delta responses following cognitive stimulation may serve as a general electrophysiological marker of cognitive dysfunction including MCI and AD [74]. The present findings add to this line of research showing that the patterns of slow EEG fluctuations, as characterized by VG features, reflect neural/cognitive abnormalities in AD. Specific to this word repetition paradigm, Xia et al. [77] has shown that the vast majority of the memory-related P600 word repetition effect is mediated by slow oscillations in the delta band. Modulation of alpha band power, in comparison, is associated with semantic processing of congruous and incongruous words. Alpha suppression was found to be greater for New than for Old words [10]. The P600 (delta activity) and alpha suppression effects reflect different aspects of verbal memory processing, and each uniquely contributes to predicting individual verbal memory performance [77].

An interesting finding in the present study is that the Old Congruous condition (words that are semantically congruous to the preceding category statements on repeated trials) produces the highest number of features. Our previous ERP studies and many behavioral studies have shown that old words are processed very differently from new words in normal elderly, due to their intact memory function, but much less so in AD patients. EEG channels producing the highest number of features were Fz, F8, R41, Pz, Br, Wl and O1. In the PCA comparision in Figure 4 and Table 4, we see this trend continue across even the different comparisons (all classes, pAD vs. RNE, AD vs. RNE). Several of these channels are known to be sensitive to word repetition and congruity manipulations in pAD patients. For example, the N400 brain potential usually becomes smaller when an incongruous word is repeated, i.e., the N400 repetition effect, and the effect is typically largest over midline and right posterior channels including Cz, Pz, Wr, R41 and T6 [7,8,13]. The P600 ERP usually becomes smaller when a congruous word is repeated, i.e., the P600 congruous repetition effect, and the effect is widespread and largest over the midline channels with a peak typically near Pz [7,8,13]. These ERP repetition effects are consistently found to be reduced or abnormal in MCI patients [7,8], and severely diminished in AD patients [13] compared to RNE, although they still appear in our comparison. The consistency across studies in channel locations where group differences were found suggests that the VG features may capture the underlying brain mechanisms related to the ERP repetition effects.

We now list strengths and limitations of our study. One of the strengths of our study is our 100% accuracy with all classifiers on AD vs. RNE which demonstrates the effectiveness of the features our method extracts. Linear separability after PCA implies that, even before dimension reduction, AD vs. RNE is still a linearly separable comparison; indeed, Figure 4 explicitly demonstrates this. Additionally, classification accuracy of 92.5% on pAD vs. RNE with non-linear neural networks and similar accuracies with linear classifiers using *only the features extracted from AD vs. RNE* highlights how these generalizable features alone may be sufficient for high-accuracy, near-linear classification of these two groups that remains competitive with other EEG-based published work which explicitly extract features for pAD vs. RNE classification [60,61,62,63,64,65,66]. This strength also likely comes from looking at word repetition EEG tasks which have been shown to be sensitive to detecting MCI-to-AD conversion and preclinical AD using ERPs [7,8,11]. Furthermore, our code is open source and linked in the paper so that future work can build upon our strong results and apply it to other datasets and tasks.

A potential limitation of the present study is the down-sampling procedure used for data reduction. Averaging EEG data across non-overlapping 80 ms time windows is effectively similar to lowpass filtering the data to 12.5 Hz, which would have reduced the amount of information in higher frequencies including beta band and above. This procedure most likely limited our ability to find discriminative VG features in these higher frequency bands. It is also worth noting that, in the present study, we used EEG time series averaged across trials for VG conversion. Cross-trial averaging is commonly used in ERP analyses to increase the signal-to-noise ratio in EEG data and extract activity that is evoked by, and phase-locked to, experimental stimuli. This averaging procedure, although highly effective as demonstrated in the present study, ignores EEG activity that is related, but not phase-locked, to the stimuli. With greater computing power, it would be valuable for future studies to identify discriminative VG features from higher frequency bands and non-phase-locked activity.

Another limitation is the small sample size used in our classification tests, feature extraction and statistical analysis (15 AD, 15 pAD, 11 RNE). We mitigate this issue in two ways: (1) we report classification scores as an average of 100 trials of training on 85% of the data and testing on 15% (and only reporting the testing accuracy), and (2) we verify the feature extraction step by only using AD vs. RNE features to classify pAD patients, demonstrating generalization of those features. Despite this, further replication of these results on larger datasets would be beneficial to the field. In such studies, it could be useful to perform data augmentation, reduce model bias by imposing some penalties during training (e.g., weight decay, dropout, etc.) or try different network architectures (such as graph neural networks) to achieve even better generalization results. An additional limitation is that we did not require amyloid biomarker studies in the definition of our clinically defined subject groups, who were well-characterized by expert clinicians and longitudinal cognitive testing.

In summary, this paper extends the results of prior studies on the use of visibility graphs for finding distinguishing features between and classifying Alzheimer’s and RNE groups [15,16] to word repetition tasks on both AD and pAD with a novel set of features. Distinguishing between pAD and RNE groups has historically produced poorer classification accuracy in the literature; however, this paper provides novel features for this type of classification that discriminates between pAD and RNE with competitive accuracy on our dataset (92.5%) simply by generalizing AD vs. RNE features. Although we achieve perfect 100% accuracy on the AD vs. RNE task and demonstrate its generalization, a larger study with a much larger sample size is still required to verify the efficacy of our framework. Because all of the code is open source, this experiment can be readily applied to much larger datasets; future applications could include predictors of conversion in MCI and discriminate between different dementia pathologies. In future work, we plan to apply our framework to larger AD and MCI datasets, and also to test similar frameworks in preclinical AD.

## Figures and Tables

**Figure 1 brainsci-13-00770-f001:**
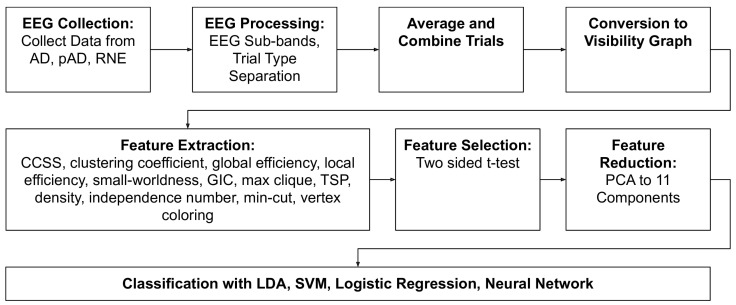
Flowchart of entire analytic process.

**Figure 2 brainsci-13-00770-f002:**
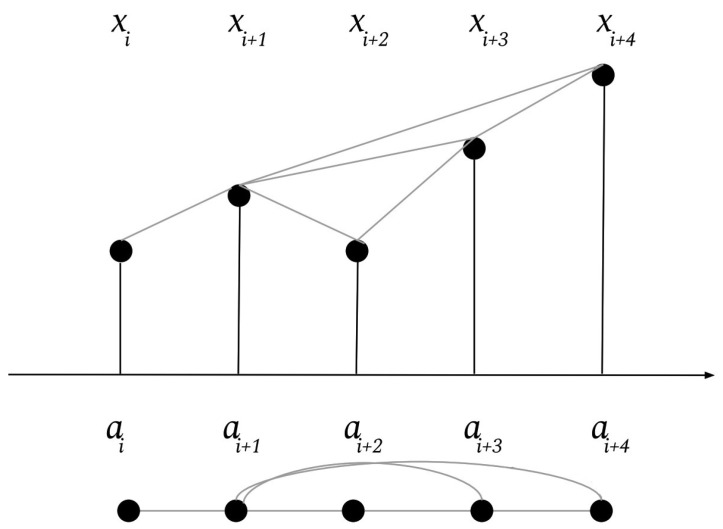
The top graph represents a time series and the edges between points signify which points can see each other. The bottom graph represents the VG of the time series, with nodes corresponding to timepoints and edges corresponding to lines of visibility. $\chi_{i}$ EEG Voltage at time *i*, $a_{i}$ VG node for timepoint *i*.

**Figure 3 brainsci-13-00770-f003:**
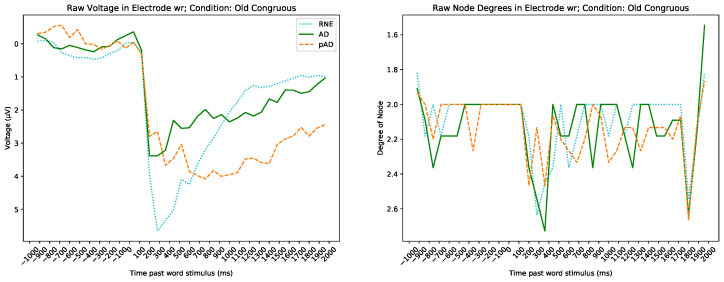
The raw band averaged voltage time series on the left, corresponding averaged degree sequences on the right. Each timepoint (represented by a unique node in the VG) in the left graph is represented by the average EEG voltage for each group; in the right graph it is represented by the average degree for its associated node in each group.

**Figure 4 brainsci-13-00770-f004:**
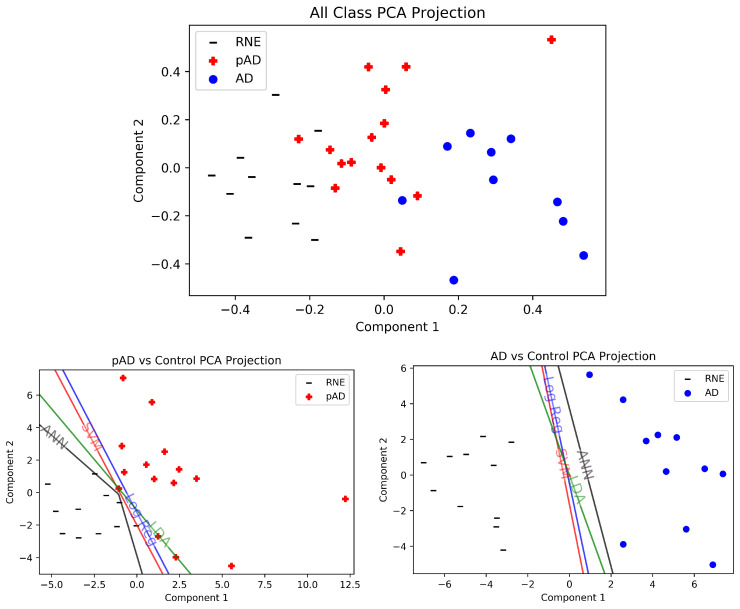
Two-dimensional PCA projections of data with associated decision boundaries for all classifiers, datapoints and comparisons. For each plot, the PCA components were computed with only the data in the plot to see the actual input to the ML algorithms. **Top:** Comparison of all classes, using the 72 extracted features from AD vs. RNE. The datapoints from all classes are projected together down to 2 dimensions for a 3-way comparison. pAD patients are intermediate to the RNE and AD patients, as expected. **Bottom Left:** pAD and RNE are also distinctly separated, resulting in excellent performance in our results by all classifiers. **Bottom Right:** In two dimensions, we easily see that AD and RNE are linearly separable with the features we extracted.

**Table 1 brainsci-13-00770-t001:** Mean ± SD values of demographics and MMSE (Mini Mental State Examination) scores in the three groups. Note: 3 AD patients had no MMSE scores; Montreal Cognitive Assessment (MoCA) scores were converted to MMSE [30] for these 3 patients.

	RNE	pAD	AD
N	11	15	11
Age (yrs)	74.1±6.8	74.6±6.9	78.5±7.5
Sex	7F, 4M	5F, 10M	4F, 7M
Education (yrs)	15.8±2.8	16.8±2.8	14.6±2.6
MMSE	29.7±0.5	26.9±2.0 *	22.9±2.8 ${}^{\#}$

* p<0.05: RNE vs. pAD, ${}^{\#}\;p<0.05$: pAD vs. AD.

**Table 2 brainsci-13-00770-t002:** Comparing the number of features produced by each band. Global Table Key: AN: All New, NC: New Congruous, NI: New Incongruous, AO: All Old, OC: Old Congruous, OI: Old Incongruous.

Category	AN	NC	NI	AO	OC	OI	Total
raw	2	2	0	2	3	3	12
delta δ	0	1	5	6	7	2	21
theta θ	2	0	0	0	1	0	3
alpha α	0	11	0	3	4	0	18
beta β	0	0	5	0	4	1	10
gamma γ	0	3	0	4	1	0	8
Total	4	18	5	19	20	6	72

**Table 3 brainsci-13-00770-t003:** Number of features produced by each channel for all conditions.

Channel	AN	NC	NI	AO	OC	OI	Total
Fz	0	7	0	3	0	0	10
Pz	0	2	0	1	2	2	7
Cz	0	0	0	0	1	1	2
F7	1	0	0	0	1	0	2
F8	0	1	4	0	5	0	10
Bl	0	1	0	0	0	0	1
Br	2	0	1	0	4	0	7
L41	0	0	0	1	0	0	1
R41	0	4	0	1	1	2	8
Wl	0	0	0	6	0	0	6
Wr	1	1	0	1	0	1	4
T5	0	1	0	1	0	0	2
T6	0	0	0	2	1	0	3
O1	0	0	0	1	4	0	5
O2	0	1	0	2	1	0	4

**Table 4 brainsci-13-00770-t004:** Top 10 PCA Loading Table. The top 10 magnitude features for each of the two components and comparisons in Figure 4. Any feature–band–electrode combination that was shared across at least two of these comparisons is bolded and shares the same superscript number in the table. The α and δ bands exclusively produce combinations shared across multiple PCA components in each comparison, and the features that appear the most across those shared are global efficiency, density, TSP and GIC. The conditions and electrodes that produce these shared features are Old Congruous, New Incongruous and New Congruous in electrodes F8, Fz, O1 and R41.

All Class	Electrode/Condition	Magnitude
Component 1	**NI: F8: Global Efficiency** δ^1^	0.191
	**NC: Fz: Max Clique** α^2^	0.178
	**NI: F8: Density** δ^3^	0.173
	**NC: Fz: Clustering Coeff** α^4^	0.168
	**NC: Fz: Local Efficiency** α^5^	0.168
	NC: Fz: Small Worldness α	0.168
	OC: Br: Local Efficiency β	0.166
	OC: Br: Clustering Coeff β	0.166
	**NI: F8: TSP** δ^6^	0.160
	OC: Br: Small Worldness β	0.156
Component 2	**OC: F8: Density** δ^7^	0.278
	**OC: F8: GIC** δ^8^	0.278
	**OC: F8: TSP** δ^9^	0.251
	**OC: F8: Global Efficiency** δ^10^	0.250
	OI: Pz: Global Efficiency Raw	0.232
	NC: Fz: Min Cut Size γ	0.210
	**OC: O1: Density** α^11^	0.209
	**OI: R41: Global Efficiency** δ^12^	0.202
	NC: O2: Density δ	0.189
	**OC: O1: GIC** α^13^	0.187
pAD vs. RNE	**Electrode/Condition**	**Magnitude**
Component 1	OC: O2: TSP α	0.209
	**NI: F8: Density** δ^3^	0.205
	AN: Wr: Global Efficiency θ	0.203
	**NI: F8: Global Efficiency** δ^1^	0.201
	NI: F8: GIC δ	0.196
	AO: Fz: Max Clique δ	0.192
	**NI: F8: TSP** δ^6^	0.186
	**OC: O1: Density** α^11^	0.183
	**OC: O1: GIC** α^13^	0.182
	**OI: R41: Global Efficiency** δ^12^	0.182
Component 2	**OC: F8: TSP** δ^9^	0.246
	OC: R41: TSP δ	0.222
	**OC: F8: GIC** δ^8^	0.208
	**OC: F8: Density** δ^7^	0.207
	OC: F8: Max Clique δ	0.206
	NC: Pz: Density γ	0.176
	AO: Fz: Clustering Coeff δ	0.170
	AO: Fz: Local Efficiency δ	0.170
	NC: R41: GIC α	0.165
	**OI: R41: GIC** δ^14^	0.162
AD vs. RNE	**Electrode/Condition**	**Magnitude**
Component 1	**NI: F8: Global Efficiency** δ^1^	0.160
	**NC: Fz: Max Clique** α^2^	0.154
	AO: L41: Max Clique α	0.152
	OC: T6: TSP γ	0.151
	**OC: F8: Density** δ^7^	0.151
	OC: O1: Global Efficiency α	0.146
	**NI: F8: Density** δ^3^	0.144
	**NC: Fz: Local Efficiency** α^5^	0.144
	**NC: Fz: Clustering Coeff** α^4^	0.144
	**OI: R41: GIC** δ^14^	0.143
Component 2	AO: Wl: Clustering Coeff β	0.271
	AO: Wl: Local Efficiency β	0.271
	AO: Wl: Small Worldness β	0.252
	**OC: F8: Global Efficiency** δ^10^	0.230
	AO: O1: Independence Number α	0.221
	**OC: F8: Density** δ^7^	0.209
	**OC: F8: GIC** δ^8^	0.206
	NC: Wr: Independence Number Raw	0.205
	OC: F7: Global Efficiency Raw	0.200
	AO: Wl: Independence Number β	0.191

**Table 5 brainsci-13-00770-t005:** Classification Statistics. Mean classification statistics and standard deviations for all classifiers on both classification tasks. The rounded best performance across each column for each classification type is bolded. AUC = Area Under (the ROC) Curve.

Type	Classifier	Accuracy (%)	Precision	Recall	AUC
AD vs. RNE	Logistic Regression	**100** ± 0.00	**1.00** ± 0.00	**1.00** ± 0.00	**1.00** ± 0.00
	SVM	**100** ± 0.00	**1.00** ± 0.00	**1.00** ± 0.00	**1.00** ± 0.00
	LDA	**100** ± 0.00	**1.00** ± 0.00	**1.00** ± 0.00	**1.00** ± 0.00
	ANN	**100** ± 0.00	**1.00** ± 0.00	**1.00** ± 0.00	**1.00** ± 0.00
pAD vs. RNE	Logistic Regression	88.50 ± 12.9	**1.00** ± 0.00	0.80 ± 0.22	**0.99** ± 0.04
	SVM	87.50 ± 13.4	0.95 ± 0.14	0.82 ± 0.20	0.95 ± 0.08
	LDA	91.50 ± 13.1	0.98 ± 0.10	0.86 ± 0.20	0.97 ± 0.06
	ANN	**92.50** ± 12.5	0.99 ± 0.07	**0.88** ± 0.20	**0.99** ± 0.05

## Data Availability

The datasets presented in this article are not readily available because they may contain identifying information and are used with permissions from the Alzheimer’s Disease Research Centers at the University of California, San Diego and the University of California, Davis. Requests to access the datasets should be directed to John Olichney, M.D.

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
