# Peer review of "Machine Learning on Visibility Graph Features Discriminates the Cognitive Event-Related Potentials of Patients with Early Alzheimer’s Disease from Healthy Aging"

_brainsci, 2023, doi:10.3390/brainsci13050770_

Round 1
Reviewer 1 Report
This study introduced a framework for electroencephalography (EEG)-based classification between patients with Alzheimer’s Disease and robust normal elderly via a graph theory approach using visibility graphs. Twelve graph features were tested for differences between the AD and RNE groups, and t-tests employed for feature selection. The selected features were then tested for classification using traditional machine learning and deep learning algorithms. Though the presented work is interesting, some critical comments need to be addressed for improving the manuscript.
1. Some important information about the experimental study should be clarified. Please give more details about (1) how many samples were used for model training and how many for testing? (2) What type of cross-validation was implemented to evaluate the experimental performance? I would suggest using a K-fold (e.g., 10-fold) cross-validation. (3) What is the dimensionality of the feature vector?
2. How varying the number of graph features would affect the classification performance?
3. I believe that it will make this paper stronger if the authors present some insightful implications based on their experimental outcomes. It would be helpful if the authors can provide an interpretation of the relationship between the learned feature representation and the pathology of the disease.
4. Some closely related studies have recently been reported. The authors need to give more review of those studies, such as: Topological network analysis of early Alzheimer’s disease based on resting-state EEG; A survey on deep learning based non-invasive brain signals: recent advances and new frontiers; Motor imagery decoding in the presence of distraction using graph sequence neural networks.
5. The authors may briefly discuss the potential limitations of the proposed method and what are the future research directions of this study. How other researchers can work on your study to continue this line of research?
Author Response
RESPONSE:
We thank you for the detailed and thoughtful review, and we are happy that you believe the work is interesting. We have made many changes to our paper based on your comments, please see our updated manuscript PDF with in-text changes highlighted in blue. We respond to individual review points and questions below:
How many samples for training/testing? What type of cross-validation—K-Fold? Dimensionality of feature vector?
We thank the reviewer for the K-fold cross-validation suggestion. We added 8-fold cross-validation results in the appendix in Table A1 and referenced these results in Section 5.2 and a new Appendix A section. The results are similar to the previous results, with all classifiers again achieving 100% accuracy on the AD vs. RNE comparison and the neural network achieving even better performance on the pAD vs. RNE comparison with an accuracy of 95.83%.
Initially, we performed cross-validation through random train-test split shuffling and reported test accuracy averaged over 100 random splits where we trained on 85% of the patients and tested on 15%. Since we have 26 total patients, 22 are in our training set, and 4 are in the testing set. We have made this clearer in 5.2.
The dimensionality of the feature vector is 11 (see Section 3.1).
How varying the number of graph features would affect the classification performance?
We have added Table A2 to the paper in the appendix with 8-Fold cross-validation results using only the 6 features introduced in prior work. Across the board, accuracies dropped on the independent pAD vs. RNE comparison. The artificial neural network’s (ANN) performance dropped the most, from 95.83% to 80.21%, indicating that the 6 features from prior work may be easier to overfit as the ANN is the only non-linear classifier and also the most expressive model class with the most learnable parameters.
This comparison verifies that the 6 novel features we introduce are important and that reducing the number of graph features extracted using only those from prior work results in worse generalization performance for all models. We have added a reference to this comparison in the main text in Section 5.2.
Analyze the relationship between the learned feature representation and the pathology of the disease?
We agree that the paper would be stronger with interpretations of the relationships between the learned features and AD pathology. We have added the following paragraphs to the discussion (Section 6):
Learned graph features, representing group differences in the morphology of EEG time series, may reflect AD pathological changes in the neural generators of ERPs, including N400 and P600. Putative N400 generators have been found in the anterior fusiform gyri and other temporal cortical regions (McCarthy et al., 1995; Halgren et al., 2022). The primary neural generators of the P600 word repetition effect were localized by functional MRI to the hippocampus, parahippocampal gyrus, cingulate, left inferior parietal cortex, and inferior frontal gyrus (Olichney et al., 2010a, 2010b). Extended synaptic failure in these regions due to AD pathology may account for the N400 and P600 abnormalities in AD and prodromal AD patients. For example, abnormal memory-related P600 potentials may be associated with tau load in the medial temporal lobe (MTL), including the hippocampus, entorhinal, and perirhinal cortices, based on the evidence that early tau accumulation in these regions correlates with lower memory performance and reductions in functional connectivity between the MTL and cortical memory systems (Berron et al., 2021).
Specific to this word repetition paradigm, Xia et al., 2020 has shown that the vast majority of the memory-related P600 word repetition effect is mediated by slow oscillations in the delta band. Modulation of alpha band power, in comparison, is associated with semantic processing of congruous and incongruous words. Alpha suppression was found to be greater for New than Old words (Mazaheri et al 2018). The P600 (delta activity) and alpha suppression effects reflect different aspects of verbal memory processing, and each uniquely contributed to predict individual verbal memory scores (Xia et al 2020).
McCarthy G, Nobre AC, Bentin S, Spencer DD. Language-related field potentials in the anterior-medial temporal lobe: I. Intracranial distribution and neural generators. J Neurosci. 1995;15(2):1080-9.
Halgren E, Dhond RP, Christensen N, et al. N400-like magnetoencephalography responses modulated by semantic context, word frequency, and lexical class in sentences. NeuroImage. 2002;17(3):1101-16.
Olichney JM, Taylor JR, Hillert DG, et al. fMRI congruous word repetition effects reflect memory variability in normal elderly. Neurobiol Aging, 2010;31(11):1975-1990.
Olichney JM, Taylor JR, Chan S, et al. fMRI responses to words repeated in a congruous semantic context are abnormal in mild Alzheimer’s disease. Neuropsychologia. 2010;48:2476-2487.
Berron D, Vogel JW, Insel PS, et al. Early stages of tau pathology and its associations with functional connectivity, atrophy and memory. Brain. 2021;16:awab114.
Mazaheri A, Segaert K, Olichney JM, Yang J, Niu Y, Shapiro K, Bowman H. EEG oscillations during word processing predict MCI conversion to Alzheimer’s disease. NeuroImage: Clinical. 2018. DOI: 10.1016/j.nicl.2017.10.009.
Xia J, Mazaheri A, Segaert K, Salmon DP, Harvey D, Shapiro K, Kutas M, Olichney JM. Event-related potential and EEG oscillatory predictors of verbal memory in mild cognitive impairment. Brain Commun. (2020). 10;2(2):fcaa213. doi: 10.1093/braincomms/fcaa213.
Cite suggested related work?
We have added the following paragraph to our introduction using the references you provided and additional studies within those references:
“Other studies which have applied neural networks or other machine learning algorithms to resting state EEG in AD include Morabito et al. (2016), who used Convolutional Neural Networks on 19 channel EEG and achieved a 3-class AD/MCI/Cognitively Normal classification accuracy of ~82% (Zhang et al. 2021, Morabito et al. 2016). Zhao and He (2014) combined Deep Belief Networks (DBN)-RBM with SVMs on 16 channel EEG signals and achieved ~92% accuracy classifying AD vs. CN (Zhang et al. 2021, Zhao and He 2014). Duan et al. (2020) quantified between-channel connectivity of resting-state EEG signals in MCI and mild AD patients using coherence measures; they used the Resnet-18 model (He. et al 2015) to classify between MCI and controls and AD and controls with an average 93% and 98.5% accuracy respectively.“
We have also added an additional paragraph on machine learning in the clinical diagnosis of AD following this paragraph: “Despite the promise of the above studies and other machine learning algorithms which have…”
Morabito F.C., Campolo M, Ieracitano C, Ebadi JM, Bonanno L, Bramanti A, Desalvo S, Mammone N, Bramanti P. Deep convolutional neural networks for classification of mild cognitive impaired and Alzheimer’s disease patients from scalp EEG recordings. RTSI 2016.
Zhang X, Yao L, Wang X, Monaghan J, McAlpine D, Zhang Y. A survey on deep learning-based non-invasive brain signals: recent advances and new frontiers. Journal of Neural Engineering 2021.
Zhao Y, He L. Deep learning in the EEG Diagnosis of Alzheimer’s Disease. ACCV 2014 Workshops.
Duao F, Huang Z, Sun Z, Zhang Y, Zhao Q, Cichocki A, Yang Z, Sole-Casals, J. Topological Network Analysis of Early Alzheimer’s Disease Based on Resting-State EEG. IEEE Transactions on Neural Systems and Rehabilitation Engineering 2020.
Limitations of the proposed method and what are the future research directions of this study? How other researchers can work on your study to continue this line of research?
The 2nd and 3rd from last paragraphs of the discussion section of the original manuscript discuss two potential limitations: (1) the down-sampling procedure used for data reduction that may limit the ability to find discriminative features visibility graph features in higher frequency bands, and (2) the small sample size (15 AD, 15 pAD, 11 RNE) that is mitigated with random sampling-based cross-validation (and now 8-fold cross validation per your suggestion) and using AD vs RNE features to classify a completely independent pAD patient group with high accuracy.
We have now reorganized the discussion section so that the strengths and limitations of our method are now labeled more clearly.
Other researchers can work on our study by building off our open-source code and using the same EEG data available upon request to the PI (we include a data availability statement at the end of the manuscript). Future works can apply our classification pipeline and novel VG features to new, independent datasets of patients with AD or prodromal AD and also test on patients with preclinical AD. We have expanded our discussion of future work in the final paragraph of the discussion.
Reviewer 2 Report
The authors present an interesting study on the use of visibility graph properties for classifying EEG recordings into patients with Alzheimer’s Disease (AD), mild cognitive impairment (MCI) and healthy elderlies. They apply a suite of pre-selected graph features to visibility graphs constructed from raw EEG recordings as well as five characteristic frequency bands, characterize their individual capacity to classify between the different groups, and finally combine them to employ four different classification algorithms based on a lower-dimensional linear subspace of the individually significant parameters. The results provide compelling evidence for a very good classification accuracy and further support the potentials of time series networks’ properties as biomarkers in clinical studies.
The present manuscript is written in a clear and concise manner. While each element of the employed methodology as a standalone appears sound, I however feel that some clarifications would further improve this work. Below, I provide a list of a few main comments, plus a couple of technical suggestions, which could be addressed in a minor-to-moderate revision of this paper.
Main comments:
1. The authors employ machine learning techniques to visibility graph features obtained from EEG recordings. This is an interesting and quite novel approach that I do like a lot. However, the way it is performed here appears a bit inconsequent to me: Machine learning is actually only applied in that last step (classification task) of the analysis to enhance the classification performance beyond what can be achieved by linear multivariate statistical methods (LDA and logistic regression). The previous, and possibly equally important step of feature selection is performed in a very classical manner by employing individual t-tests between groups to each of the (heuristically pre-selected) graph properties. I do not want to request the authors to completely revise their study design for the presented work. However, I feel that they should elaborate a bit on the question why they do not make use of any of the powerful machine learning alternatives to the feature selection (e.g. random forests) and maybe also the dimensionality reduction (beyond PCA) steps as well. From the results obtained by the authors, it seems like such more sophisticated methods would most likely not bring any further improvements (given that the classification accuracy with the used approach is already close to 100%), but I feel the justification of the used methodology as it stands now a bit incomplete.
2. The authors claim to use a wavelet decomposition to separate the different frequency bands for each of the recorded EEG channels. However, what they describe in Section 2.3 is ordinary bandpass filtering. This should be clarified.
3. In lines 141-143, the authors emphasize that they are generating averaged time series, but the actual procedure (and use) of this averaging is hard to grasp. In essence, what has actually been done does only become clear in Section 5, so considerably later. I recommend enriching this part of the text with some further details.
4. Lines 174-177: The way this sentence is phrased may suggest that all mentioned properties have been employed to EEG studies using visibility graphs, while at least one of the cited references used functional networks of multi-channel EEGs (as does the CCSS measure defined in Section 2.6.1). I suggest to be more specific in this aspect.
5. In general, the pre-selection of the precise 12 visibility graph based features used in this manuscript is not fully obvious. I understand that it is quite hard to go through an exhaustive list of all possible graph properties and explain why specific ones are (not) selected, but some more general ideas on this point could be interesting. Specifically, as the authors rightly point out, some of the properties characterize similar graph features and may hence not be fully independent of each other. Although such dependences seem to have been minimized in the selected suite of approaches (e.g. by not considering some standard measures like closeness or harmonic centrality as node-based characteristics), I feel that this point should be addressed somewhere in the paper. (Giving an overall correlation matrix between all 12 features, maybe except for CCSS, might be one option, but this is just a suggestion and not a request.)
6. If I have counted correctly, 5 of the 12 used graph characteristics (size of max clique, TSP cost, independence number, size of minimum cut and vertex coloring number) take integer values by definition, while the others are rational. Can this have an effect on the classification performance (or, more precisely, the amount of information provided by each of the 12 characteristics to the overall classification, i.e. explanation of the inherent group structure of the data)? Related to this point: By performing a PCA on the space of significant features prior to employing different classifiers on a lower-dimensional subspace, a possible attribution of classification skill to specific graph characteristics is hampered. (Accordingly, I also wonder how feature importance for prediction is being “measured” in line 398, since this appears not quite straightforward to me in the proposed analysis setup.) I suppose that for practical purposes, one might be interested in estimating just a lower number of graph features that contain in essence similar information on the different EEG categories. My suggestion to the authors would be discussing the aforementioned aspects at some appropriate point of their manuscript.
7. Regarding the feature extraction step: Could you please clarify the use of two-tailed t-tests for assessing the significance of differences of individual graph properties between groups? (You report to study three groups, but use a test for differences between pairs of groups.) What is the rationale for not employing other techniques tailored for the latter setting (e.g. one-way ANOVA or nonparametric versions thereof)? In a somewhat related manner: employing a critical p-value of 0.01 is a rather “hard” criterion (in many studies, 0.05 is used instead) – is this choice somehow related to the fact that graph features and also different time series (raw vs. filtered) are not independent, so that the probability of false positives may in fact be larger than 0.01? (This would also affect some of the statements in Section 5.1.)
8. The AVERAGED degree sequences shown in Fig. 3 might be considered a quite meaningless property (and have, accordingly, also not been used by the authors for their classification task), since a lot of relevant information might get lost by the averaging. Notably, the left panel also shows a common feature (edge effect) with systematically reduced degree at the first and last nodes of a sequence. (Why is the direction of the y axis reverted in the corresponding plot?) Notably, except for the first and last node, the minimum degree of each node in a visibility graph is 2 by definition (connection to nearest neighbors), so the fact that the authors commonly did not observe higher values suggests a quite “boring topology”…
Technical comments:
- I feel that complete affiliation information in the manuscript header should be standard.
- Line 11: Please define the abbreviation MCI at its first occurrence in the abstract (not just in l.27).
- Figure 1: I think that “Neural Network” in the classification step should be specified.
- Fig. 2: Figure text (“legend”) and caption overlap in the PDF, please check.
- Equation (1): please use a single line for the equation and separate the conditions j\in\mathbb{Z}_+ and j- Section 2.6.1: I would add a brief note clarifying that this property employs the functional network concept, while visibility graph properties are used as similarity criteria to establish the latter.
- The concept in Section 2.6.1 makes use of the local clustering coefficient of nodes, which is however only introduced below in Section 2.6.2. I suggest integrating the definition of C_i into Section 2.6.1 and hence shortening Section 2.6.2.
- Table 4: please enlarge font size
- In quite a few references, journal names have not been properly capitalized. Moreover, references 33 (page numbers/article ID?) and 40/46 (article ID instead of artificial page numbers) should be revised/completed.
Author Response
Response:
We thank you for the very detailed review and are happy that you believe the results are compelling. We have made many changes to our paper based on your comments, please see our updated manuscript PDF with in-text changes highlighted in blue. We respond to individual review points and questions below:
- ML is applied at the last step. Why not use more powerful ML feature selection and dimensionality reduction methods?
In early experiments, we did try more sophisticated methods, such as recurrent neural networks directly on the EEG signals. However, these end-to-end methods immediately overfit to the training inputs despite applying different forms of model regularization, highlighting the need to extract features from very noisy signals like EEG recordings. This was a sign that very powerful models trained without first performing feature selection would perform poorly on our data. Furthermore, in the EEG for AD classification literature, more powerful techniques for feature selection, like random forests and dimensionality reduction beyond PCA, do not seem to be commonly used, perhaps due to the issues we discovered in the early experiments. It is possible that with very large patient datasets, such techniques may be used, as it becomes harder to overfit to such datasets, but typical EEG AD datasets are similar in size to the ones we evaluate on and are likely to be too small to use more powerful techniques while maintaining good generalization performance.
- Section 2.3: Wavelet Decomposition in title but not used.
Thank you for pointing out this error. We did not perform wavelet decomposition on the datasets used for classification (only explored this in a few individual’s data) and have removed this from the section title. Rather, standard bandpass filtering was used.
- Better explain how authors are generating averaged time series?
We have clarified this now (in Section 2.4) with: “For every patient, we obtained multiple word repetition trials for each experimental condition. To reduce the noise in the EEG signal and extract event-related information, we average across trials in each condition so that there is one averaged EEG time series per condition, frequency band, and channel combination.”
- L174-177: The way this sentence is phrased may suggest that all mentioned properties have been employed to EEG studies using visibility graphs, while at least one of the cited references used functional networks of multi-channel EEGs.
We have clarified this in the text by adding citations to each of the individual features we mention to highlight which specific graph features were studied in the corresponding cited works. See Section 2.6: “Six of these features have been tested…namely clustering coefficient sequence similarity [15], average clustering coefficient [28, 29], global efficiency [16, 28, 30]...”
- Selection of the 12 VG features is not obvious.
We agree that the selection of the features is not exactly obvious. Six of the features selected are present in prior work and have been shown to be useful for high-accuracy AD classification. To better motivate the six novel features we introduce, we have added the following sentence to Section 2.6 before going over all of the features in detail: “In general, the new features we introduce come from classic, well-studied problems in graph theory and are targeted towards extracting information specifically about VG structure (e.g., visibility of vertices induced by the time series structure).”
- Some graph features are integer values, some are rational. Does this affect performance?
We do not believe the fact that some values are integers while others are rational has an effect on classification performance. This is because before applying PCA, we utilized the common practice to first normalize the data so that the PCA process can extract directions of high explained variance of the data across features of different scale. After normalization and before PCA, integer-valued features are thus scaled to a mean 0 distribution taking on the property of rational numbers.
- PCA makes it hard to attribute classification performance to specific features. Why not use less VG features overall?
We agree that performing PCA before employing classifiers makes it harder to directly attribute classification importance to various graph features. However, due to the expected number of false positives after t-testing, we believe that this direct attribution may not be too meaningful. Thus instead, we display a PCA loading table in Table 4 with associated analysis in Section 5.1 and in the discussion, where we examine the magnitude (eigenvalue) of the top 10 PCA features (eigenvectors) for a two-component PCA projection. This comparison highlights the importance of specific graph features in specific band-channel combinations for classification while ensuring that features that may not be important, even after t-testing, are not examined.
Per your suggestion, we have now included a comparison of using only the 6 features from prior work (none of our 6 novel features) for classification in Appendix A. Across the board, accuracies dropped on the independent pAD vs RNE comparison. The artificial neural network’s (ANN) performance dropped the most, indicating that the 6 features from prior work may be easier to overfit to as the ANN is the only non-linear classifier and also the most expressive model class with the most learnable parameters.
This comparison verifies that the 6 novel features we introduce are important and that reducing the number of graph features extracted using only those from prior work results in worse generalization performance for all models. We have added a reference to this comparison in the main text in Section 5.2.
- Why t-tailed and not ANOVA if looking at 3 groups?
We only extract features from the AD vs RNE comparison (see Section 2.6: “In total, we used 12 features to classify the AD and RNE groups)---our pAD vs RNE classification task is performed only with features extracted from AD vs RNE patients and therefore demonstrates the generalization capabilities of these features. Hence there is no need to use a one-way ANOVA test that is generally designed for 3 or more groups (Mishra et al. 2019).
Mishra P, Singh U, Pandey CM, Mishra P, Pandey G. Application of student's t-test, analysis of variance, and covariance. Ann Card Anaesth. 2019 Oct-Dec;22(4):407-411. doi: 10.4103/aca.ACA_94_19. PMID: 31621677; PMCID: PMC6813708.
- Why p-value 0.01 and not 0.05?
We use the p-value of 0.01 simply to reduce the number of possible false positive features as we are testing the statistical significance of a total of 5796 predictors.
- Averaged sequences in Fig 3 are not that meaningful? Why is the y-axis reversed in 2nd plot?
We used the averaged degree sequences in Fig. 3 because individual patient EEG signal sequences can vary a lot, therefore making a cross-group (RNE vs pAD vs AD) visual comparison difficult. This figure is meant to serve simply as an illustrative example of general trends that can be seen in the conversion of EEG signals to VGs. The direction of the y axis is reverted in the corresponding node degree plot to keep the direction of the y axis consistent (larger positive values plotted downward) in both plots for better clarity and consistency.
Technical comments:
Thank you, we have made these changes.
Reviewer 3 Report
This paper presents graph features using a different classifier to detect and classify Alzheimer’s disease prediction.
-Introduction part is not written clearly. Please rewrite the section with more proper references.- More literature review is required. Add more recent references related topic.
-Introduction section is very short. More description with overall prediction framework.
-The methodology section is poorly written. -Your accuracy is average accuracy or it is only for one iteration?
-Give limitations and benefits.
-Give future works
- What are your novelties and contributions? Why do the proposed visible graph features give better results than other recent algorithms? Give proper justification.
-In Table 5 the accuracy of AD vs RNE is 100%. It looks bias result. Check the machine learning framework.
-Which cross-validation strategy is used k-fold or LOOCV?
-Your proposed approach is limited to one database only. Verify your result with another database.

Author Response
We thank you for the thoughtful review. We have made many changes to our paper based on your comments, please see our updated manuscript PDF with in-text changes highlighted in blue. We respond to individual review points and questions below:
Moderate English Changes Required
We have made many changes throughout (see tracked changes throughout the updated manuscript) fixing minor clarity issues.
Introduction needs more references, a literature review, and description of framework?
Thank you for the suggestion. We have added 3 additional paragraphs to the introduction:
- The first paragraph gives an overview of other studies which have used EEGs for AD classifications without visibility graphs.
- Another paragraph explains the current “gold standard” for the clinical diagnosis of AD and that machine learning algorithms hold promise in improving diagnostic accuracy, but have not yet been widely applied.
- The third paragraph gives an overview of our prediction framework.
Throughout these paragraphs, we have added many more references to recent related works.
Methodology section: average accuracy or only for one iteration?
We report average accuracies on the testing set (see Section 5.2, “We report classification metrics on the 15% testing set, where the metrics are averaged across all 100 trials for each model”).
Give limitations and benefits.
The 2nd and 3rd from last paragraphs of the discussion section of the original manuscript discuss two potential limitations: (1) the down-sampling procedure used for data reduction that may limit the ability to find discriminative features visibility graph features in higher frequency bands, and (2) the small sample size (15 AD, 15 pAD, 11 RNE) that is mitigated with random sampling-based cross-validation (and new 8-fold cross-validation results in Appendix A) and using AD vs. RNE features to classify a completely independent pAD patient group with high accuracy.
The 4th from last paragraph of the original manuscript discusses some strengths: (1) 100% accuracy with all classifiers on AD vs RNE and linear separability of the data from the two classes in this comparison, and (2) high classification accuracy on the independent pAD vs RNE comparison with a non-linear model (the neural network) comparable to prior EEG-based work that explicitly extracted features for the pAD vs RNE classification task.
We have now reworded these paragraphs so that the strengths and limitations of our method are now labeled more clearly.
Future work?
Future work can apply our classification pipeline and novel VG features to new, independent datasets of patients with AD or prodromal AD and also to patients with preclinical AD. We have expanded our discussion of future work in the final paragraph of the discussion.
What are your novelties and contributions? Why do the proposed features give better results than other algorithms?
Most prior work in the ERP field has focused mainly on extracting pre-defined components (e.g., N400, P600) of the EEG signal for analysis, which reduces the complexity of the overall data and ignores many intricacies of the recorded EEG time series themselves. Some works have tried converting resting-state EEG signals to visibility graphs for feature extraction, analysis, and classification, which has demonstrated some promising results. Our work here is novel. It utilizes recorded EEG data during a word repetition task, which has been shown to be very sensitive to detecting AD (Olichney et al. 2006 [13], Olichney et al. 2008 [8]), combined with a set of 12 visibility graph features (6 of which are novel) to extract local and global information within individual EEG recording time series and then uses these for high-performance classification. We demonstrate that these features are generalizable and use them to detect earlier stage AD (pAD) with very high accuracy through comprehensive comparisons across many machine learning algorithms. Another contribution is that we open-source our code so other researchers can build upon it in future work.
We have modified parts of the last two paragraphs of the introduction to make these contributions clearer.
While it is very difficult to compare algorithms done in different AD and elderly samples, most of the other algorithms discussed used resting state EEG, rather than cognitive ERPs, which are based on an average of many single trials of a mental task and therefore greatly reduces variability. Furthermore, the particular (word repetition) paradigm employed may confer additional advantages as it was designed to test memory and semantic processing, abnormalities of which are considered “cardinal features” of early AD.
In Table 5 the accuracy of AD vs RNE is 100%. It looks bias result. Check the machine learning framework.
The ML framework has been checked and verified. We achieve 100% discriminability also with the K-fold (K=8) cross-validation method, see the new appendix section. This is explained by our feature extraction pipeline’s ability to extract relevant features such that the AD vs RNE classification problem is linearly separable (see Figure 4, bottom right).
What cross validation is used?
Initially, we performed cross-validation through random train-test split shuffling and reported test accuracy averaged over 100 random splits where train on 85% of the patients and test on 15%. However, we now have added k-fold (k=8) cross-validation results in appendix Table A1 and referenced these results in Section 5.2 and a new Appendix A section. The results are similar to the previous results, with all classifiers again achieving 100% accuracy on the AD vs RNE comparison and the neural network achieving even better performance on the pAD vs RNE comparison with an accuracy of 95.83%.
Verify with an additional database?
Although a limited-sized sample (n=15), the prodromal AD patients that we test on constitute a second (independent) dataset, and classification accuracy remained very high (92.5% to 95.8%). We further highlight that we used the features extracted from the AD patients on the prodromal AD classification task, verifying that the features studied by our pipeline can generalize.
It is our future plan to apply this model to larger AD & MCI datasets and also to test VG models in preclinical AD,as we have added to last paragraph of the Discussion.
Reviewer 4 Report
The use of artificial intelligence for early diagnosis of dementia and the tracking of cognitive, non-pharmacological interventions preventing the development of senile dementia is very much developed for the sake of social good. Presented research results for all classifiers on both classification tasks have high precision and can translate into further refinement of algorithms. Below are some comments and suggestions:
1. In the introduction it should be included information on which artificial intelligence algorithms are most often used in the diagnosis of diseases associated with memory loss in the elderly;
2. In the subsection 2.5 it should be written what are the limitations of Visibility Graphs - whether the accuracy of the result plays a role in the evaluation;
3. Subsections 2.6.6, it should be added two/three sentences, in particular about the characteristics and importance for patients;
4. Subsections 2.6.10, 2.6.11, 2.6.12 I propose to combine into one subsection due to the fact that they do not contain significant formulas;
5. In table 2, it should be added full names for: Greek letters and abbreviations: AN, NC, NI, AO, OC, OI. In addition, the chapter is too short to be a standalone chapter, I suggest expanding it to add diagrams, drawings for the principal component analysis;
6. The fourth chapter should be significantly expanded with characteristics for the methods: support vector machines; linear discriminant analysis and fully connected artificial neural network - in particular with regard to their advantages;
7. The third Figure, should be enlarged - so that the unit every 100 ms is marked on the horizontal axis;
8. One statement should be added to the conclusions regarding the results of the tests depending on the number of patients.
Author Response
We thank you for the detailed and thoughtful review, and we are happy that you believe the work is aligned with social good. We have made many changes to our paper based on your comments, please see our updated manuscript PDF with in-text changes highlighted in blue. We respond to individual review points and questions below:
Intro: Add info on which AI algorithms are most often used in diagnoses of memory diseases.
Following your advice, we have added two additional paragraphs in the introduction that give an overview of recent work on using machine learning and EEG for accurate detection of Alzheimer’s Disease and the current “gold standard” for clinical diagnosis. See the paragraphs starting with: “Despite the promise of the above studies and other machine learning algorithms…” and its new preceding paragraph.
We have also added current references summarizing the promise of artificial intelligence algorithms for diagnosing neurodegenerative diseases in these two paragraphs (Zhang et al. 2021, Tautan et al. 2021). However, to date, there is not yet a widely applied AI algorithm for the clinical diagnosis of AD (see Dubois et al. 2021 [24]).
Section 2.5: What are VG limitations?
Per your suggestion, we have added the following text to Section 2.5: “In general, VGs are biased towards creating local edges that capture information about the signal over short periods of time, with the exception of peaks in the signal. VGs can also only be extracted per electrode, however, we compensate for this by extracting a cross-channel feature, as detailed below.”
Section 2.6.6: Add info about characteristics and importance for patients.
We agree that it would be better to explain the graph index complexity intuitively. We have added the following to 2.6.6: “This eigenvalue lies somewhere between the average and max node degree. Therefore, a larger GIC may correspond to a more complex signal structure resulting from, for example, more frequent signal voltage fluctuations.”
Table 2: Full names for greek letters and abbreviations.
Thank you, we have added full names for the greek letters and added a table key explaining the abbreviations.
Section 3: Expand section on PCA.
We have expanded section 3 per your suggestion by adding a paragraph explaining exactly how principal component analysis is performed and its derivation from the eigendecomposition of the normalized covariance matrix of the data.
Section 4 should be expanded with information about each of the ML classifiers.
We have added the following detailed comparison of the methods in Section 4:
- Logistic regression: logistic regression is a commonly used, simple classifier that learns a linear decision boundary by learning a single feature vector through gradient descent.
- Support vector machines: SVMs learn a decision boundary with a “margin” away from datapoints from either class that is maximized. This can result in better testing error as the decision boundary should not lie too close to points of either class.
- Linear discriminant analysis: LDA models the data distributions of both classes as Gaussians with equal covariances and draws a linear decision boundary between the means of the two Gaussians. LDA can perform very well if the input data follows these assumptions.
- Artificial neural network: the ANN can linear non-linear decision boundaries in the feature space of the input data. It has the potential to overfit more easily to the data but also to learn better-fit decision boundaries if the true decision boundary must be non-linear.
Figure 3: Should have marks every 100 ms.
Thank you for the suggestion, we have made the change to Figure 3.
One statement should be added to the conclusions regarding the results of the tests depending on the number of patients.
Our discussion contains a paragraph on limitations, stating: “Another limitation is the small sample size used in our classification tests, feature extraction, and statistical analysis (15 AD, 15 pAD, 11 RNE). We mitigate this issue in two ways:...”
We have now reorganized the discussion section so that the strengths and limitations of our method are now labeled more clearly.
Round 2
Reviewer 1 Report
The authors have addressed my concerns. The revision is looking good and can be considered for publication.
Author Response
We thank the reviewer for their response!
Reviewer 3 Report
All comments given by reviewers are addressed by authors. I am satisfy with the revision made by authors. Now it is ready for publication.
Author Response

(The authors gave the same response as above.)
